



# Coupled effects of observation and parameter uncertainty on urban groundwater infrastructure decisions

Marina RL Mautner[1,2], Laura Foglia[1], and Jonathan D Herman[2]

[1]Department of Land Air and Water Resources, University of California Davis, Davis, CA, USA
[2]Department of Civil and Environmental Engineering, University of California Davis, Davis, CA, USA
**Correspondence:** Marina RL Mautner (mmautner@ucdavis.edu)

**Abstract.** Urban groundwater management requires complex environmental models to represent interactions between hydrogeological processes and infrastructure systems. While the impacts of external uncertainties have been widely studied, there is limited understanding of how decision support is altered by endogenous uncertainties arising from model parameters and observations used for calibration. This study investigates (1) the importance of observation choice and parameter values on
aquifer management objectives when controlling for model error, and (2) how the relative performance of management alternatives varies when exposed to endogenous uncertainties, individually and in combination. We use a spatially distributed groundwater model of the Valley of Mexico, where aquifer management alternatives include demand management, targeted infiltration, and wastewater reuse. The effects of uncertainty are evaluated using global sensitivity analysis, performance ranking of alternatives under a range of human-natural parameters, and identification of behavioral parameter sets filtered with an error
metric calculated from varying subsets of observations. Results show that the parameters governing hydraulic conductivity and total water use in the basin have the greatest effect on management objectives. Model errors are not necessarily controlled by the same parameters as the objectives needed for decision-making. Additionally, observational and parameter uncertainty each play a larger role in objective variation than the management alternatives themselves. Finally, coupled endogenous uncertainties have amplifying effects on decision-making, leading to larger variations in the ranking of management alternatives than
each on their own. This study highlights how the uncertain parameters of a physically-based model and their interactions with uncertain observations can affect water supply planning decisions in densely populated urban areas.

## 1 Introduction

Groundwater resource planning and management requires increasingly complex models to represent interactions between hydrogeological and infrastructure systems to achieve sustainability (Megdal et al., 2015; Singh, 2014; Wada et al., 2017;
Peters-Lidard et al., 2017). A key challenge for model-based decision support is understanding the influence of multiple sources of uncertainty on the choice of infrastructure alternatives. In particular, the role of external uncertainties such as future climate, population, and land use change, have been investigated extensively in the systems analysis field (Hadka et al., 2015; Maier et al., 2016; Kwakkel and Haasnoot, 2019). Similar approaches have been applied in groundwater systems to analyze the combined effects of perturbations in external forcing (Dams et al., 2008, 2012; Mustafa et al., 2019; Fletcher et al., 2019).





However, the endogenous uncertainties arising from physically-based hydrologic and hydrogeologic models are often neglected in infrastructure planning studies, despite often influencing predictions as much or more than external drivers (Mendoza et al., 2016; Qiu et al., 2019; Herman et al., 2020).

Physically-based groundwater models can support infrastructure decisions by ranking alternatives according to their performance under stakeholder-defined management objectives. Global sensitivity analyses of the ranking of alternatives have

generally focused on the influence of objective values and weights in multi-criteria decision models, without providing a physical basis for the determination of such variations (Hyde and Maier, 2006; Ganji et al., 2016). As a result, these decision models often do not account for uncertainty in hydrologic processes, leaving an opportunity to relate processes to the criteria values that are produced for a given management alternative. For example, Ravalico et al. (2009, 2010) analyze the effects of parameter changes in on the optimal policy ranking by determining the minimum, median, and maximum parameter values

that change the ranking of alternatives based on a single management objective; however, their implementation did not address model error. Specifically, none of the existing approaches explicitly evaluates the relationship between uncertain endogenous model characteristics used to determine model error and ranking of management alternatives for decision-making based on model output.

In hydrogeologic models, endogenous uncertainty is contributed by model parameters describing natural and human com-

ponents of the system, the set of historical observations used to calibrate or constrain the parameters (Moore and Doherty, 2005; Doherty and Simmons, 2013). Parameters provide the flexibility to represent complex systems on a broader scale, and in some cases can encapsulate differences in model structure as well (Guillaume et al., 2016). The propagation and attribution of parameter uncertainty has been the topic of numerous hydrologic modeling studies, using a combination of uncertainty analysis and sensitivity analysis (Razavi et al., 2021; Pianosi et al., 2016), though generally without considering the influence of this

uncertainty on model-based decision support (Jing et al., 2019), or only focusing on local sensitivity analyses (Tolley et al., 2019). Global sensitivity analysis in particular has seen growing usage with advances in computing power (Razavi and Gupta, 2015), including sensitivity varying over time and/or space (Herman et al., 2013; Şalap-Ayça and Jankowski, 2016; Reinecke et al., 2019; Zhang and Liu, 2021), and model structure (Mai et al., 2020). Observational uncertainty is typically also excluded, except in the case of inverse modeling (Refsgaard et al., 2007).

The choice of observations to support parameter identification is often complicated by a number of factors, including: temporal and spatial representation of the model area, data quantity and quality, and resolution of datasets that determine model structure with respect to observation locations (McMillan et al., 2018; Refsgaard et al., 2012; Lehr and Lischeid, 2020). This is especially true for groundwater modeling in urban environments, where infrastructure, monitoring practices, and pumping patterns can complicate groundwater data collection procedures meant to ensure accurate and repeatable results (Foster et al.,

1998; Vázquez-Suñé et al., 2010; Bhaskar et al., 2016). Uncertainty in the selection of observations will alter the parameter calibration (Montanari and Di Baldassarre, 2013), and in turn, the planning problem (Brunner et al., 2012). For example, Etter et al. (2018) found that the inclusion of simulated uncertain observations in calibration had a negative impact on the performance of streamflow models, but that the effect was mitigated with a larger sample size of observations. Similarly, Rojas





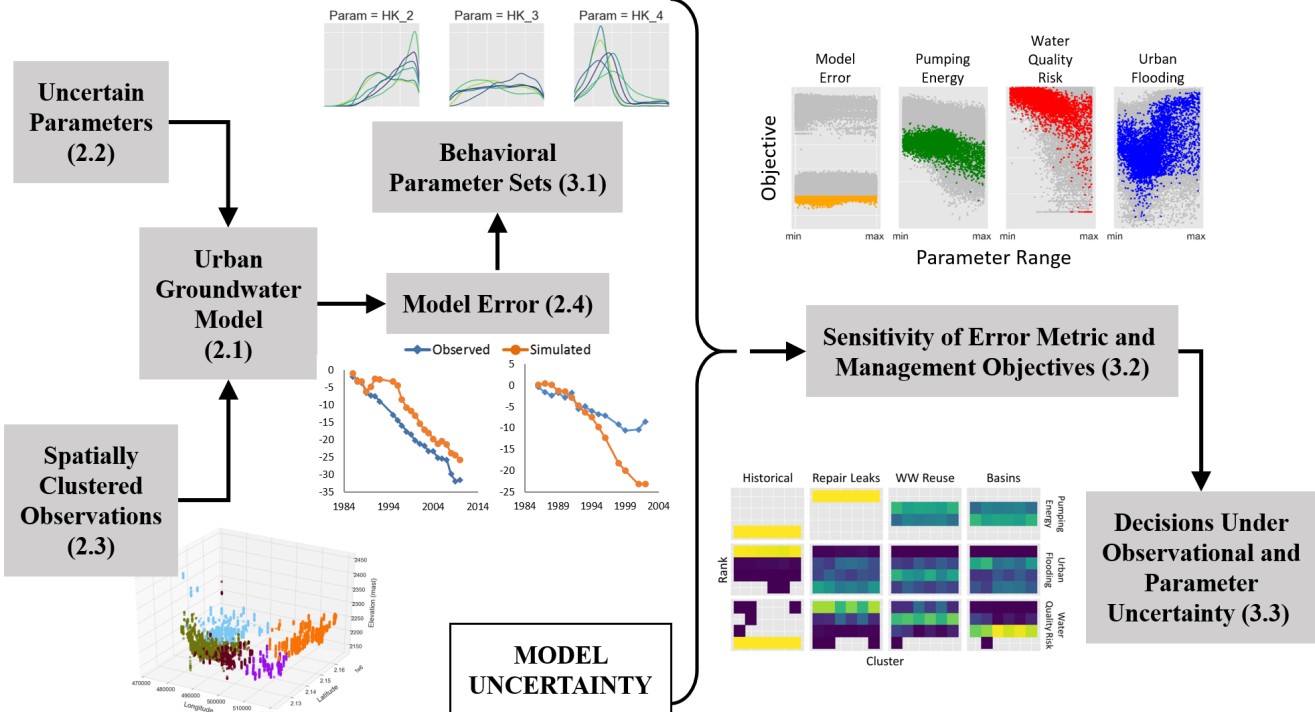

**Figure 1.** Flowchart of methods.

et al. (2010) explore the availability and variety of observations in characterising the choice of conceptual models in multimodel
analysis, again focusing on effects on model error.

When developing groundwater models for planning purposes, calibration is often carried out by selecting a "best" parameter
set by minimizing one or more error metrics while adjusting parameter values, using parameter sensitivity or expert evaluation
to determine which parameters to adjust. Alternatively, some calibration frameworks use observations and the resulting behav-
ioral model space of a selected error metric to refine the distribution of parameter values, rather than optimizing a single one
(Wagener et al., 2003; Bárdossy, 2007; Beven, 2016). In such calibration frameworks, a behavioral parameter set comprises
a sample of parameter sets from the behavioral model space through minimization of the error metric. A number of studies
have focused on improving behavioral parameter set analysis by including regional datasets and expert knowledge in addition
to parameters and inputs (Kelleher et al., 2017) or evaluating sets that perform poorly with respect to a given error metric
in addition to acceptable simulations (Reusser and Zehe, 2011). However, beyond prior studies of model error, there remains
a need to understand the coupled effect of uncertainty in hydrogeologic model parameters and observations on the relative
performance of decision alternatives (Razavi et al., 2021).

This study aims to evaluate the sensitivity of groundwater model error and decision-relevant management objectives to
uncertain parameters and observations, and to determine the effects of this coupled uncertainty on the infrastructure planning





problem. The result is a planning-driven evaluation of uncertainty to support groundwater management, with the goals of

identifying parameters to improve the accuracy of the hydrogeologic model as well as those that should be better constrained

to support the selection of management alternatives. This is done through a combination of a global sensitivity analysis and

a performance ranking under a range of human-natural parameters, and with the identification of behavioral parameter sets

based on multiple possible subsets of historical observations (Figure 1). These diagnostic methods aim to evaluate two main

consequences of these decision-relevant uncertainties: first, the importance of observation choice and parameter values on the

absolute objective performance when controlling for model error, and second, how the relative performance of management

alternatives varies when exposed to endogenous uncertainties, individually and in combination. This approach exemplifies how

the propagation of multiple endogenous uncertainties throughout the modeling process can ultimately affect the outcomes of

regional groundwater supply planning.

## 2   Methodology

This study focuses on the Mexico City Metropolitan Area to evaluate the effects of parameter and observation uncertainty on

multi-objective groundwater modeling and decision-making. The Mexico City Metropolitan Area lies within the southwestern

portion of the Valley of Mexico watershed, characterized by volcanic peaks surrounding a high plains basin (OCAVM, 2014).

This paper uses a case study of the urban aquifer management problem in the Valley of Mexico using a spatially distributed

groundwater model adapted from prior work (Herrera-Zamarrón et al., 2005; Lopez-Alvis, 2014; Galán-Breth, 2019; Mautner

et al., 2020). This type of complex three-dimensional model is required to approximate the interactions between physical

hydrogeologic properties and managed aquifer recharge interventions. This model complexity makes uncertainty analysis

difficult, but also critical to understand how spatially and temporally aggregated management objectives vary across many

parameter combinations.

### 2.1   Urban Groundwater Model

The Valley of Mexico model is written in Python using the *flopy* package to preprocess data and run the model in MODFLOW,

a widely used software which solves the groundwater flow equation (Bakker et al., 2016), as presented in Mautner et al. (2020).

The following is a brief overview of the Valley of Mexico test case. A set of model parameters govern model representation

of geologic setting, land use and land cover, and water resource infrastructure in Mexico City, including artificial and natural

recharge, time varied groundwater pumping, and heterogeneous subsurface characteristics (Table 1). This model covers an area

of 84 km by 67 km on a 500x500 m spatial grid, and the time period from 1984 to 2013. All model inflows and outflows

are applied at a daily time-step, varied according to a monthly stress-period, meaning that data is provided at the monthly

time-scale, although data availability may cause some fluxes to vary at the annual or decadal time-scale. Four groundwater

recharge alternatives are simulated: spatially distributed infiltration basins, demand management through repair of leaks in the

water supply network, injection of treated wastewater at existing wastewater treatment plants, and the status quo historical

alternative.



**Table 1.** Model Parameters and Sampling Ranges

| Param | Units | Lower Bound | Upper Bound | Param | Units | Lower Bound | Upper Bound |
|---|---|---|---|---|---|---|---|
| **Zonal Geologic** | | | | **Time Varied Infrastructure** | | | |
| $HK_1$ | m/d | 8.64E-7 | 5.00E-2 | $Q1990$ | - | 0.3 | 2.25 |
| $HK_2$ | m/d | 1.00E-1 | 1.00E+2 | $Q2000$ | - | 0.45 | 3.5 |
| $HK_3$ | m/d | 3.46E-2 | 1.50E+2 | $Q2010$ | - | 0.5 | 4 |
| $HK_4$ | m/d | 4.32E-2 | 4.32E+1 | $LK_{1990}$ | - | 0.5 | 2 |
| $HK_5$ | m/d | 4.32E-4 | 8.64E+1 | $LK_{2000}$ | - | 0.5 | 2 |
| $S_{s,1}$ | 1/m | 9.19E-4 | 2.03E-2 | $LK_{2010}$ | - | 0.5 | 2 |
| $S_{s,2}$ | 1/m | 4.92E-5 | 1.05E-3 | $TWU_{1990}$ | - | 0.75 | 2 |
| $S_{s,3}$ | 1/m | 1.00E-7 | 6.89E-5 | $TWU_{2000}$ | - | 0.95 | 2 |
| $S_{s,4}$ | 1/m | 1.00E-7 | 1.02E-4 | $TWU_{2010}$ | - | 1.1 | 2 |
| $S_{s,5}$ | 1/m | 1.00E-7 | 6.89E-5 | | | | |
| $S_{y,1}$ | - | 0.001 | 0.08 | **Zonal Recharge** | | | |
| $S_{y,2}$ | - | 0.05 | 0.4 | $RCH_{urban}$ | % | 0 | 10 |
| $S_{y,3}$ | - | 0.01 | 0.2 | $RCH_{natural}$ | % | 1 | 80 |
| $S_{y,4}$ | - | 0.05 | 0.4 | $RCH_{water}$ | % | 10 | 50 |
| $S_{y,5}$ | - | 0.001 | 0.1 | | | | |
| $VANI_1$ | - | 1 | 1000 | **Leak Infiltration** | | | |
| $VANI_2$ | - | 1 | 1000 | $IN$ | % | 5 | 50 |
| $VANI_3$ | - | 0.1 | 100 | | | | |
| $VANI_4$ | - | 0.1 | 100 | | | | |
| $VANI_5$ | - | 0.1 | 100 | | | | |

[1]Lacustrine, [2]Alluvial, [3]Fractured basalt, [4]Volcaniclastic, [5]Andesitic

$^{HK}$Horizontal hydraulic conductivity, $^{S_s}$Specific storage, $^{S_y}$Specific yield, $^{VANI}$Vertical anisotropy of hydraulic conductivity, $^{Q}$Urban pumping multiplier, $^{LK}$Ratio of distribution leaks to estimated leaks using 1997 data, $^{TWU}$Regional water use multiplier, $^{RCH}$Recharge percentage for each land use type, $^{IN}$Infiltration percentage for leaked water

Each alternative is then evaluated according to three aquifer management objectives: pumping energy use, water quality risk, and urban flood risk (Equations 1-3). The management objectives evaluated are adapted from Mautner et al. (2020) to function under the constraints of the global sensitivity analysis performed for this study. The pumping energy objective ($E$) is governed by the energy required to pump a daily quantity of groundwater ($p$) from the water table ($h$) to the ground surface 110 ($s$) across all time periods ($t$) of varying length in days ($d$) and across all pumping wells ($w$), converted to kilowatt hours using an efficiency and conversion term ($\epsilon$). $E$ is calculated starting in the 3rd year of the model period to avoid spin-up effects. In





the Valley of Mexico, the lacustrine aquitard in the center of the valley serves as a barrier to contamination of the underlying productive alluvial aquifer; ensuring that the hydraulic head remains above the confining layer reduces water quality impacts in the long-term. The water quality risk objective ($W$) indicates the number of cells not meeting the groundwater levels below the

confining lacustrine layer necessary to maintain water quality ($l$) divided by the total number of lacustrine cells in the model ($L$) during the time periods ($t$) in the last year of the model period. The urban flood risk ($F$) is the sum of the urban area in cells with groundwater mounding ($a$) divided by the total urban area in the model ($A$) during the time periods ($t$) in the last year of the model period.

$$E = \sum_{t=25}^{360} \sum_{w=1}^{n_{wells}} \epsilon pd\left(s_{t,w} - h_{t,w}\right) \tag{1}$$

$$W = \frac{\sum_{t=348}^{360} l_t}{\sum_{t=348}^{360} L} \tag{2}$$

$$F = \frac{\sum_{t=348}^{360} a_t}{\sum_{t=348}^{360} A} \tag{3}$$

### 2.2 Uncertain Parameters

The 33 model parameters include zonal geologic, time varied infrastructure, zonal recharge, and infiltration characteristics (Table 1). There are four zonal geologic parameters for each of the five geologic formations; one parameter for each of the three decades during the model period for the total water use, ratio of urban to periurban pumping, and distribution system leak multiplier; a recharge percentage for each land use type; and an infiltration parameter for leaked water. Parameter ranges are adapted from Mautner et al. (2020), adding or adjusting maxima and minima where necessary based on literature and physical

relationships. The calibration carried out in Mautner et al. (2020) used a local sensitivity analysis in which some parameters were not assigned sampling ranges. In this study, a global sensitivity analysis is used and thus some combinations of parameter values had to be avoided based on the structure of the model (Table 1). For example, estimated pumping in the region is determined by subtracting historical non-pumping water source quantities from the total regional water use derived from the total water use multiplier [TWU], thus, the TWU must result in a regional water use greater than the historical non-pumping

water sources. Similarly, the urban pumping multiplier [Q] acts on a historical dataset, and must result in total pumping less than the estimated pumping determined by the combination of regional water use, historical other supply sources, and the TWU multiplier. The selected parameter ranges are shown in Table 1. Using these ranges, 100,000 unique parameter sets are generated using Latin Hypercube sampling. Simulations for each of the management alternatives using the parameter sets were carried out on 296 processors over a total of 107,814 CPU hours.



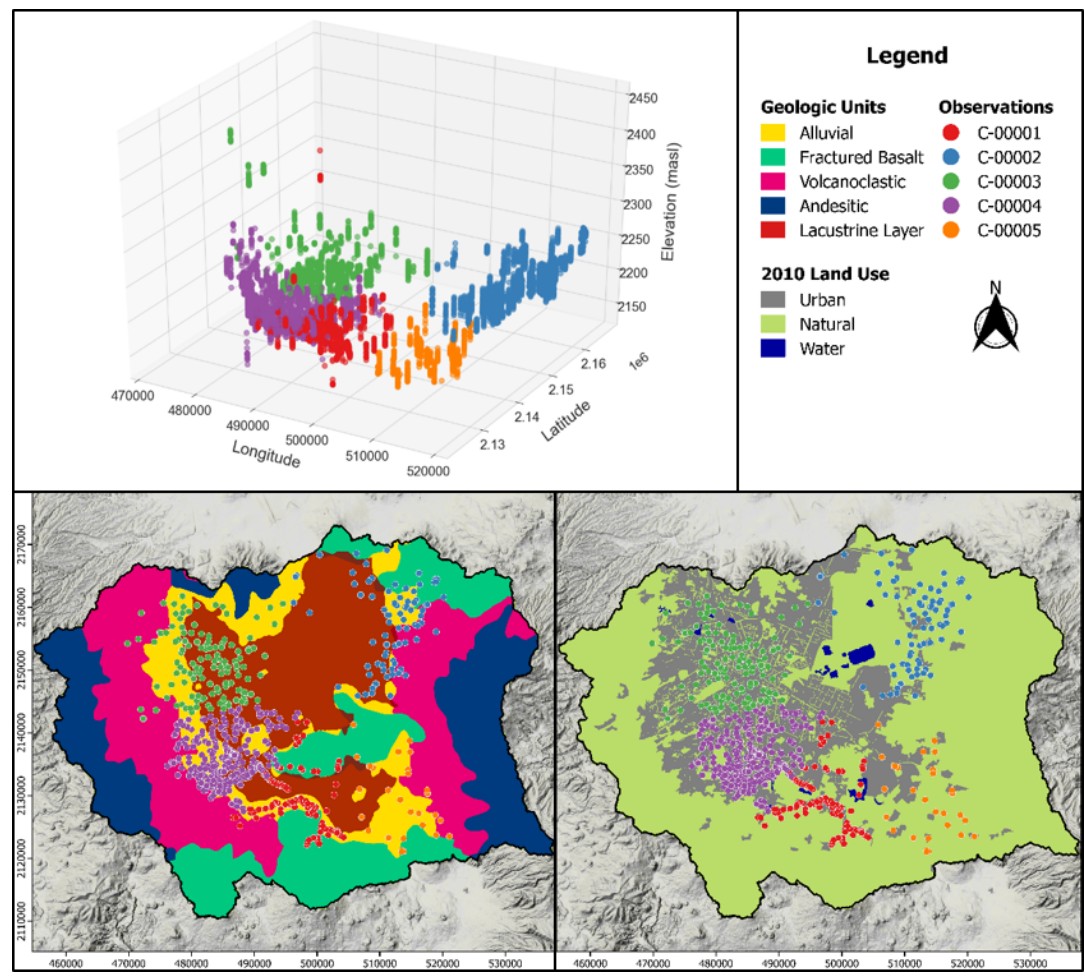

**Figure 2.** (Top) A 3-dimensional visualization of the 5 clusters of observations used in this study. (Bottom left) Observation clusters shown with the geologic formations within the model area. (Bottom right) Observation clusters shown with the land use types for the model period covering 2010.

## 2.3 Spatially Clustered Observations

Uncertainties introduced throughout the groundwater modeling process propagate through to decision-making based on the simulated performance of management alternatives. Parameter uncertainty is reduced by calibration against observations. However, there is also error in, and uneven representation of the model area by, the observations used for calibration. Ideally, observation data can be filtered according to knowledge of collection methods, characteristics of monitoring wells, and distribution across the model area. However, increasingly, modelers face unwieldy and incomplete observation data sets that have greater degrees of freedom and limited or uncertain boundary conditions with which to calibrate models (Tiedeman et al., 2004; Tonkin et al., 2007; Hrachowitz et al., 2013; Nearing et al., 2021). These uncertainties can include lack of data on geo-





logic formation boundaries, placement and magnitude of cones of influence from pumping wells, and the effects of urban karst
and land cover types on natural and artificial recharge near monitoring wells.

A set of 8,181 observations from 676 monitoring wells is available for the area and time period of the urban groundwater
model used in this study. Measurements are available at the annual time scale with 1 to 29 data points per well over the 30 year
model period, depending on the well. Multiple interacting uncertainties (e.g. land use, pumping wells, geologic formations)
can have unpredictable effects on the relevance of certain observations, thus observation uncertainty is represented in this
study by separating the full set of observations into randomly selected and spatially distinct subsets of observations to be act

as proxies for incomplete historical records (Figure 2). The full set was separated into five clusters using centroid initialization
of K-means clustering normalized within the 3-dimensional space of the set (Figure 2), resulting in a total of six clusters when
the full set is included as a control.

## 2.4   Model Error

Doherty and Moore (2020) propose the selection of a decision-critical prediction when assimilating observed data into a model

for calibration. In this model, the three groundwater planning objectives are based on various spatial and temporal aggregations
of groundwater head values, thus, an error metric that assesses model agreement with piezometric head, the decision-critical
prediction, through space and time was selected. As in Mautner et al. (2020), model error is captured by the sum of squared
weighted residuals (SSWR) between historical head observations ($h_{obs,i}$) and simulated values ($h_{sim,i}$), using weights ($\omega$)
determined in Lopez-Alvis (2014) and Galán-Breth (2019):

$$SSWR = \sum_{i=1}^{n} \frac{1}{\omega^2}(h_{obs,i} - h_{sim,i})^2 \qquad (4)$$

The model error is calculated under the status quo scenario to characterize model agreement with historical hydraulic head
observations. Given the inclusion of multiple observations for a single well over time, the error metric captures both spatial
and temporal variability in hydraulic head. Higher values of this metric indicate poor model agreement with observations, with
larger disagreements amplified in the metric as a result of the squared residual.

## 2.5   Parameter Set Selection

In complex systems with uncertain inputs, model processes can be difficult to parameterize and even more difficult to constrain.
While perfect monitoring and representation is the ideal, in reality, it is necessary to simplifying assumptions that then must be
calibrated to create models that can better inform policy and management. In such cases, it is common to have multiple viable
parameter sets that produce simulations with acceptable or equivalent model error. It is possible that changes in the observa-
tions used to evaluate error can leads to differences in the behavioral parameter sets that are chosen as the best performing

simulations. Here, we calculate the error metric for each of the six observation clusters, inclusive of the full set of observations,
and choose the 5% best performing parameter sets according to that metric. This gives a sample of 5,000 parameter sets that





perform relatively well with respect to the full sample of 100,000. We refer to these as the cluster behavioral parameter sets for each of the six observation clusters.

## 2.6 Sensitivity Analysis

To better ensure robust management alternatives under uncertain model inputs, global sensitivity analysis has been increasingly explored as a decision support tool (Razavi et al., 2021). The sensitivity of the management objectives across the parameter space both across management alternatives and cluster behavioral parameter sets indicates the variability of uncertainty with respect to individual physical model parameters. Using the cluster behavioral parameter sets, a global sensitivity analysis is performed using the Delta Moment-Independent Measure ($\delta$) Borgonovo (2007); Plischke et al. (2013). This method was selected

for two reasons. First, $\delta$ provides a better representation of sensitivity with respect to model structure when parameters are correlated, often true in complex human-natural systems (e.g. increased groundwater pumping during periods of reduced recharge and surface supplies from drought), when compared to variance-based methods (Borgonovo and Plischke, 2016). Second, the Delta method does not require a specific structure of parameter samples, allowing for the sub-selection of 5,000 samples from the initial set. By only evaluating objective sensitivity across the solution space of the cluster behavioral parameter sets rather

than the entire solution space, we remove objective values of simulations that do not agree with observations, which have the potential to introduce further uncertainties. Parameter sensitivity is calculated for four model outputs: the error metric, and three management objectives. A total of 72 sensitivity analyses on 33 parameters are performed across combinations of: 4 alternatives, 3 objectives, and 6 cluster behavioral parameter sets, resulting from filtering based on the error metric among each of the 6 observation clusters. The sensitivity analyses were performed on 72 processors over a total of 4.9 CPU hours.

## 2.7 Evaluation of Decision Uncertainty


To understand the extent to which uncertainty in observations and parameters can affect decision-making analyses, we compare alternative performance across cluster behavioral parameter sets. First, management alternatives are ranked within each objective for each of the parameter sets to view differences in the alternative ranking across the cluster behavioral parameter sets. We evaluate the model results to visualize changes in ranking according to three types of comparisons using sets of

heatmaps that present the number of parameter sets that are ranked at each level across the four alternatives. The comparisons evaluated are: all three objectives across the observation cluster behavioral sets, all three objectives across the range of one of the model parameter values (HK2, alluvial hydraulic conductivity), and the water quality objective across the observation cluster behavioral sets and the alluvial hydraulic conductivity parameter values simultaneously.

For the comparison across observation cluster behavioral sets, first, in each parameter sample the alternatives are ranked from

lowest (1) to highest (4) by the objective value. This is repeated for each of the 5,000 parameter samples in each observation cluster behavioral set and the count of samples in which the alternative is ranked at each level (1-4) is recorded. As in the previous comparison, for the comparison across parameter values, in each parameter sample the alternatives are ranked from lowest (1) to highest (4) by the objective value. This is done for the behavioral set selected using the C-12345 (all observations) cluster. Then, for each parameter, the 5,000 sample set is separated into ten bins along the entire parameter value range from





Table 1. In this comparison, the count of alternatives at each rank is divided by the total number of parameter samples in each bin to allow for direct comparison across all bins. This is necessary because the non-uniform distribution in some parameters caused by the behavioral parameter set selection would obscure the relative performance of the alternatives along the parameter distribution otherwise. For the final comparison, the method for the comparison across the parameter values is repeated for the remaining five behavioral parameter sets. These are compared for the water quality objective as an example.

Second, the difficulty of the decision was measured by evaluating the percent difference between the first and second ranked alternatives, and between the first and worst ranked alternatives. The distribution of these differences indicate the relative performance between the alternatives, with a distribution concentrated among lower values indicating a more difficult decision because the relative differences between the objective measures of the options are smaller. While alternative ranking can provide some information on the relative performance of aquifer management alternatives with respect to each other, it does

not provide information on the difference between the performance in each simulation. More importantly, by not knowing the range of objective values between the management alternatives in a given simulation, decision-makers might incorrectly infer the difficulty of a decision. For example, take the case of two simulations where the performance in the urban flood risk objective of the historical, infiltration basin, wastewater reuse, and repair leaks alternatives are 1.5%, 1.7%, 1.8%, and 2%, respectively, in the first simulation, and 2%, 15%, 32%, and 40% in the second simulation. These two simulations may produce

the same alternative ranking: historical (1), infiltration basins (2), wastewater reuse (3), and repair leaks (4). However, it is clear that second simulation produces a much "easier" decision than the first because the absolute and relative differences between the objective values are larger in the second simulation than in the first.

## 3   Results and Discussion

### 3.1   Cluster Behavioral Parameter Sets

Figure 3 shows the kernel density estimations (KDEs) for the resulting parameter distributions when selecting the 5,000 samples with the lowest error using each of the observation clusters (C-00001, C-00002, C-00003, C-00004, C-00005) and the entire observation set (C-12345). The initial distribution (not shown) is uniform for all parameters. These distributions indicate the parameters that have the greatest influence on model error, defined here as those with the greatest deviation in distribution from the prior uniform distribution, namely the horizontal hydraulic conductivity [HK] parameters. The higher parameter

values for the geologic characteristics (horizontal hydraulic conductivity [HK], specific storage [SS], and specific yield [SY]) of the alluvial formation (formation number 2) are preferentially represented in the low-error parameter sets. For hydraulic conductivity, this indicates that an alluvial formation [HK2] that allows for more rapid flow of groundwater, and thus greater dispersion of groundwater throughout the model area, results in lower error. When combined with high values for flux parameters such as the total water use [TWU] (governing groundwater pumping) and recharge [RCH], this could signal that these

models avoid extreme mounding or drawdown that would increase model error. Similarly, the selection of a larger number of high values for specific storage [SS2] and specific yield [SY2] in the alluvial formation confirms that the selected parameters would tend to mitigate the effects of higher flux values. Alternatively, all distributions of the horizontal hydraulic conductivity





**Figure 3.** The distributions along the parameter ranges of the filtered samples using the sum of squared error metric. The distributions are colored according to the observation cluster used to filter the dataset. The prior distribution (not shown) is uniform for all parameters. Parameter abbreviations given in Table 1.





show a concentration of lower values for the volcaniclastic formation [HK4]. This indicates parameter values that encourage higher groundwater retention in the mountainous, volcaniclastic areas, which could be a result of observations in perched or

mountainous regions having an outsized effect on the error metric.

In terms of flux parameters, the total water use [TWU], the recharge percentage of the natural land use type [RCH2], and the leak [LK3] and pumping [Q3] multipliers for the third decade of the model period all show a small redistribution toward the extremes of the parameter ranges. The preference for lower values of total water use, particularly in the first decade [TWU1], could confirm that mitigated drawdown in the model leads to lower error. At the same time, the slight tendency toward increased

recharge in the natural land use type agrees with the tendency toward low hydraulic conductivity in the volcaniclastic formation that, combined, would indicate a preference for groundwater mounding along the model edges. Finally, the higher values for the leak parameter in the last decade of the model period [LK3] further confirms the preference for increased hydraulic head in the urban areas.

In isolation, these findings reveal information about the model representation and how to improve parameterizations to

minimize error given the existing observations. However, there are visible differences between the distributions of the parameter values from the various cluster behavioral parameter sets. This is particularly evident in the hydraulic conductivities of the alluvial [HK2], fractured basalt [HK3], and volcaniclastic formations [HK4]. Behavioral parameter sets tend to focus on sub-ranges of the horizontal hydraulic conductivity depending on the subset of observations used to calculate the error metric, highlighting the importance of observational uncertainty on parameter identification.

Error reduction through parameter selection is an important consideration for model use. However, we are also interested in how management objectives produced by the model respond to uncertainty in model parameters. Figure 4 shows the error metric and the three management objectives for all parameter sets in gray and the behavioral parameter sets in color. Here we visualize how the choice of observation cluster affects the sample of parameter sets and subsequently the range of performance among the pumping energy (green), water quality risk (red), and urban flooding objectives (blue). This example yields noticeable

differences between the observation cluster choices, while other parameters (Figure S1) result in fairly uniform sampling across the parameter ranges, following Figure 3. The three objectives are to be minimized, thus, in certain objectives, higher alluvial hydraulic conductivity [HK2] results in better performance, particularly for the energy and water quality objectives, while in the flooding objective the performance is more variable across the parameter range. This performance is not consistent across clusters for the alluvial hydraulic conductivity, indicating the impact of observational uncertainty on the performance

evaluation of the system.

## 3.2    Sensitivity of Error Metric and Management Objectives

To better understand the effects of parameter values on management objectives, the Moment-Independent Sensitivity Measures, $\delta$, are shown for the energy objective in Figure 5 (Figure S2 for water quality risk and Figure S3 for urban flooding risk). As in Figure 4, the patterns of objective sensitivity to the parameters vary across the samples chosen using different observation

clusters. However, in Figure 5 we can also compare the sensitivity of the objectives across management alternatives. With few exceptions, the sensitivities of the objectives across the alternatives within each cluster sample are fairly consistent. This





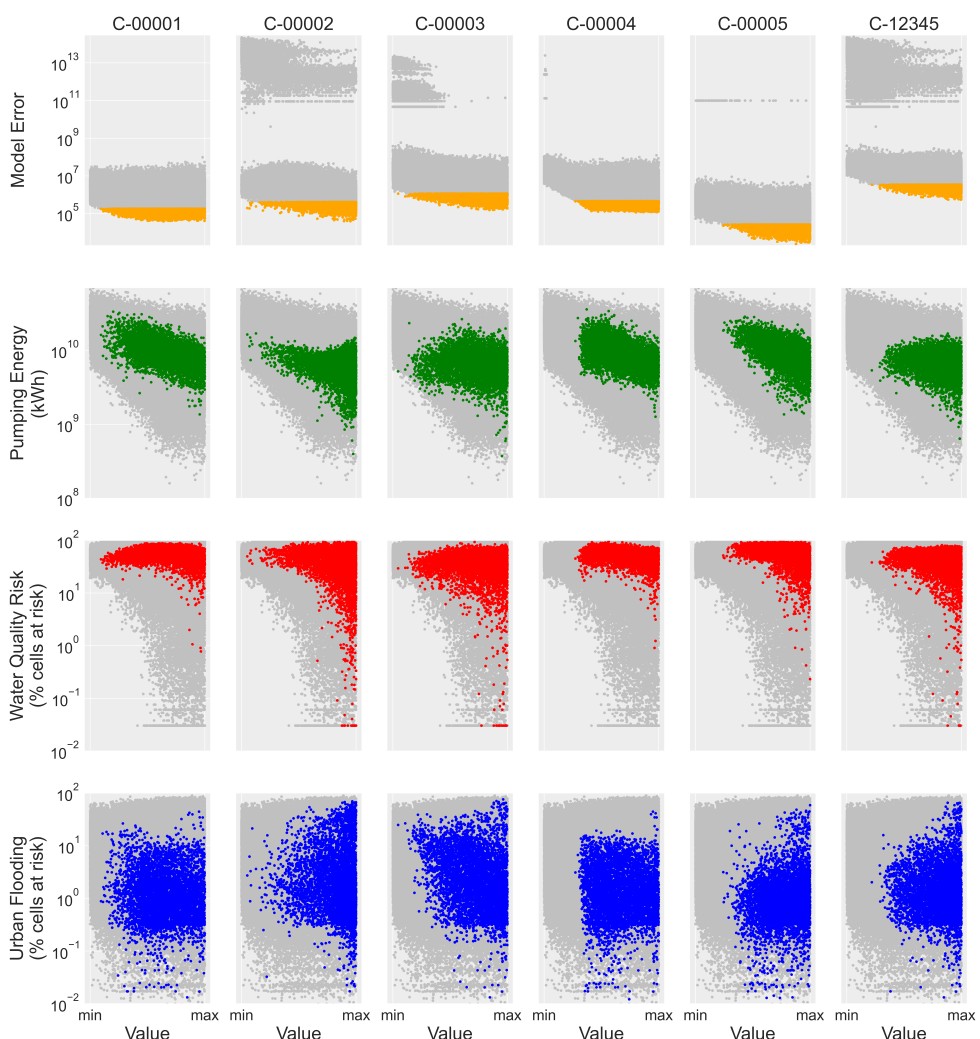

**Figure 4.** A representative view of the four model output metrics for the historical alternative, plotted against the parameter range for the hydraulic conductivity of the alluvial formation (the most sensitive parameter from Figure 3). These include the error metric (sum of squared weighted residuals), energy objective (kWh), water quality risk objective (percent of cells not meeting the objective), and urban flooding objective (percent of cells not meeting the objective). Gray points represent all parameter sets, while colors represent behavioral parameter sets meeting the error threshold.





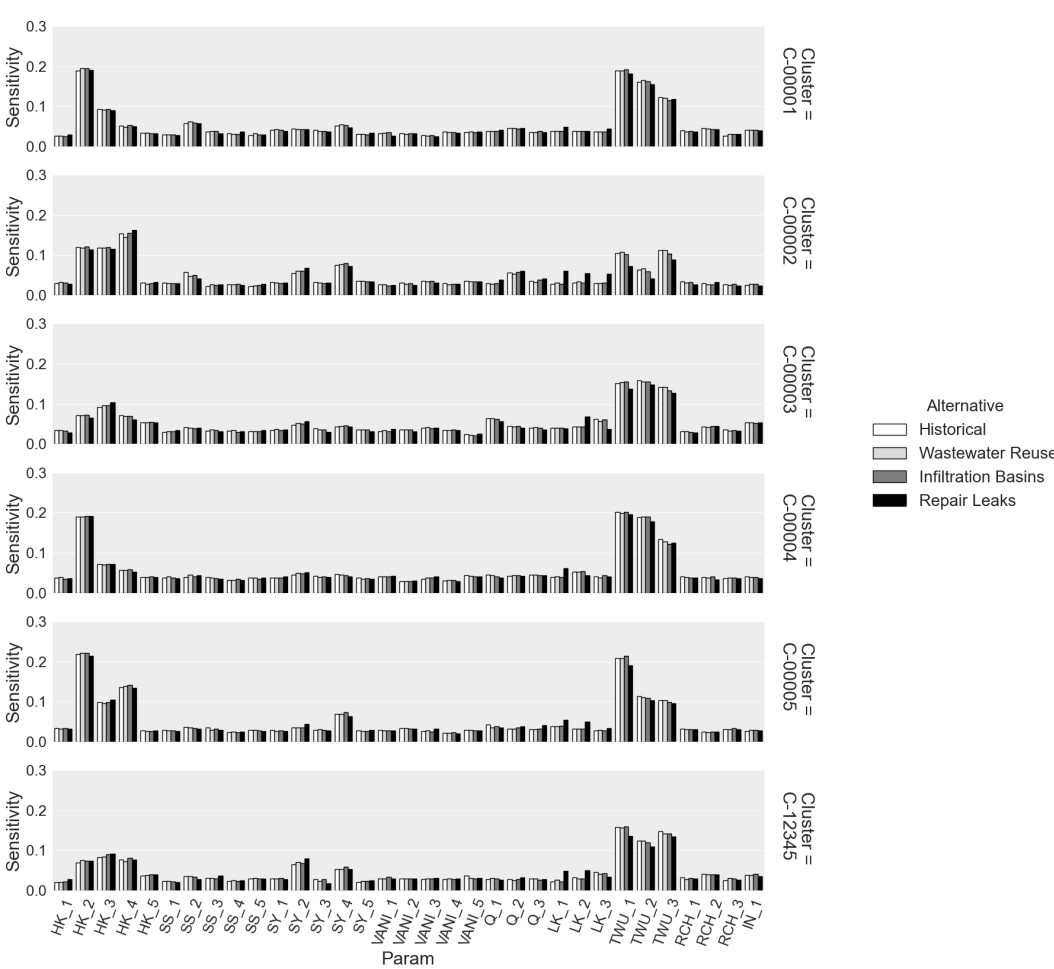

**Figure 5.** $\delta$ sensitivity of the energy objective according to the 5,000 filtered samples for the 33 model parameters (columns). The sensitivity is shown by cluster (rows) and by the four alternatives from left to right (light to dark): historical, wastewater reuse, infiltration basins, and repair leaks.

suggests that the performance of the system with respect to the management objectives is minimally affected by the choice of alternative. This has two main implications. First, this could signify that the relative performance of the alternatives is similar across a range of parameter values and indicate that the decisions made are robust across many parameter combinations. Second, if decision makers are using a sensitivity analysis to choose parameters for further study, they can be relatively confident that the choice of parameters to monitor will not favor a given alternative. The most notable exception is the sensitivity of the pumping energy objective with respect to the leak multiplier (LK1, and to a lesser extent LK2 and LK3) for the repair leaks alternative. This is expected given the reliance of the leak repair alternative on the quantity of leaks present–essentially, the





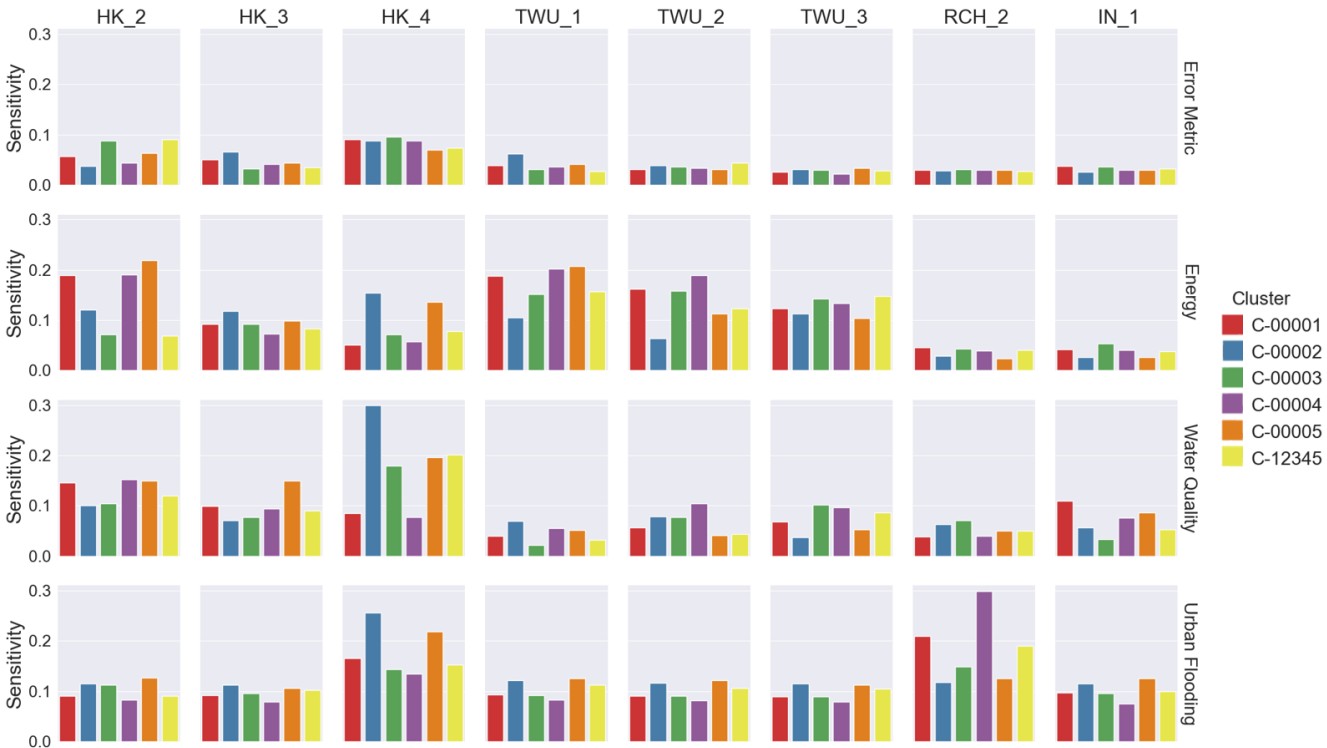

**Figure 6.** $\delta$ sensitivity of the error metric and three management objectives (rows) according to the 5,000 filtered samples for the 33 model parameters (columns) for the historical management alternative. The sensitivity is shown by cluster in order from left to right: C-00001, C-00002, C-00003, C-00004, C-00005, C-12345.

more leaks available to be repaired indicates a larger water savings and thus a higher water table from which to pump to the
ground surface.

It is also valuable to understand how the sensitivities of the three management objectives compare to those for the error metric. Many numerical groundwater models are constructed with a specific management purpose, but the model itself is calibrated to error metrics that represent available data, and these may not necessarily rely on the same mechanisms driving the performance of management alternatives. Figure 6 shows the $\delta$ values for the parameters with the largest differences in
sensitivity between clusters. The sensitivities of the error metric across the filtered sample are relatively small because they include only the parameter sets with the lowest error. While the sensitivities of the error metric to the parameters are smaller overall than those of the objective values, the effects seen on the distributions in Figure 3 are mirrored to some extent here, with slight increases in the sensitivity of the error metric to the horizontal hydraulic conductivity of the alluvial [HK2] and volcaniclastic [HK4] formations.
However, the patterns of the sensitivity of the error metric generally do not align with the patterns seen in the management objectives. Objectives are more or less sensitive to specific parameters depending on the cluster behavioral parameter sets.





For example, the sensitivity of all three management objectives to the volcaniclastic hydraulic conductivity [HK4] is largest for the C-00002 samples, and is most pronounced for the water quality risk objective. Figure 4 shows that the parameter sets selected using the observation cluster C-00002 result in a much broader set of values for the hydraulic conductivity of the volcaniclastic formation than the other objective cluster samples, particularly for the water quality risk and urban flooding indicators. Similarly, samples C-00001 and C-00004 result in much higher sensitivities of the urban flooding objective to the recharge parameter of the natural land cover [RCH2].

Higher sensitivities for certain cluster behavioral parameter sets may indicate that the chosen observations do not properly constrain the model with respect to the given parameter, resulting in a number of non-unique solutions. Alternatively, higher sensitivities may occur when the spatial extent of the parameter and the management objective calculation are coinincident, as in the case of the total water use parameters [TWU], which act upon the pumping wells, and the energy objective, which is calculated at the location of the pumping wells. Finally, the sensitivities are also affected by the physical processes governed by a given parameter, as in the case of the high sensitivity of the urban flooding objective to the recharge percentage parameter [RCH]. Understanding which parameters contribute most to objective uncertainty indicates opportunities for data collection to improve model representation of those processes. The $\delta$ values show that uncertainties in the observations used in calibration can result in appreciable changes in the distribution of the performance in management objectives. These findings underline the importance of high quality, well distributed, and diverse observation data for calibration. Additionally, decision-making often depends on the behavior of spatially and temporally aggregated indicators or objectives whose sensitivity to model parameters may or may not be aligned with the sensitivity of the error metric to those same parameters.

## 3.3 Decisions Under Observational Uncertainty

Parameter sensitivities provide information about improvements that can be made in the modeling and calibration process to reduce error. However, it is also important to understand how these uncertainties propagate into the decision-making process, particularly whether they contribute to changes in potential decisions informed by the simulation model. Figure 7 shows the relative performance of the aquifer management alternatives according to the cluster behavioral parameter set and management objective. In the heatmaps, a lighter (yellow) color indicates more parameter sets where that alternative is ranked at that value and a darker (purple) color indicates fewer parameter sets that are ranked at that values. If no parameter sets result in a given rank for that alternative, the space is left gray.

For the pumping energy objective, the historical and repair leaks alternatives rank worst (4) and best (1), respectively, across all simulations in all parameter set samples, while the wastewater reuse and infiltration basin alternatives rank 2nd and 3rd almost evenly across the simulations. The wastewater reuse alternative ranks 2nd slightly more often (lighter) in the pumping energy objective than the infiltration basin alternative, particularly in the C-00001 cluster behavioral parameter set and the full observation sample set (C-12345). In the water quality risk objective, the historical alternative ranks 4th across practically all the cluster behavioral parameter sets. Similarly, the infiltration basins alternative ranks 3rd in almost all behavioral parameter sets. The 1st and 2nd ranked alternatives, while less definitive are still fairly clear, with the repair leaks alternative ranking 1st and the wastewater reuse alternative 2nd across most of the cluster behavioral parameter sets. Here, C-00003 and C-00005

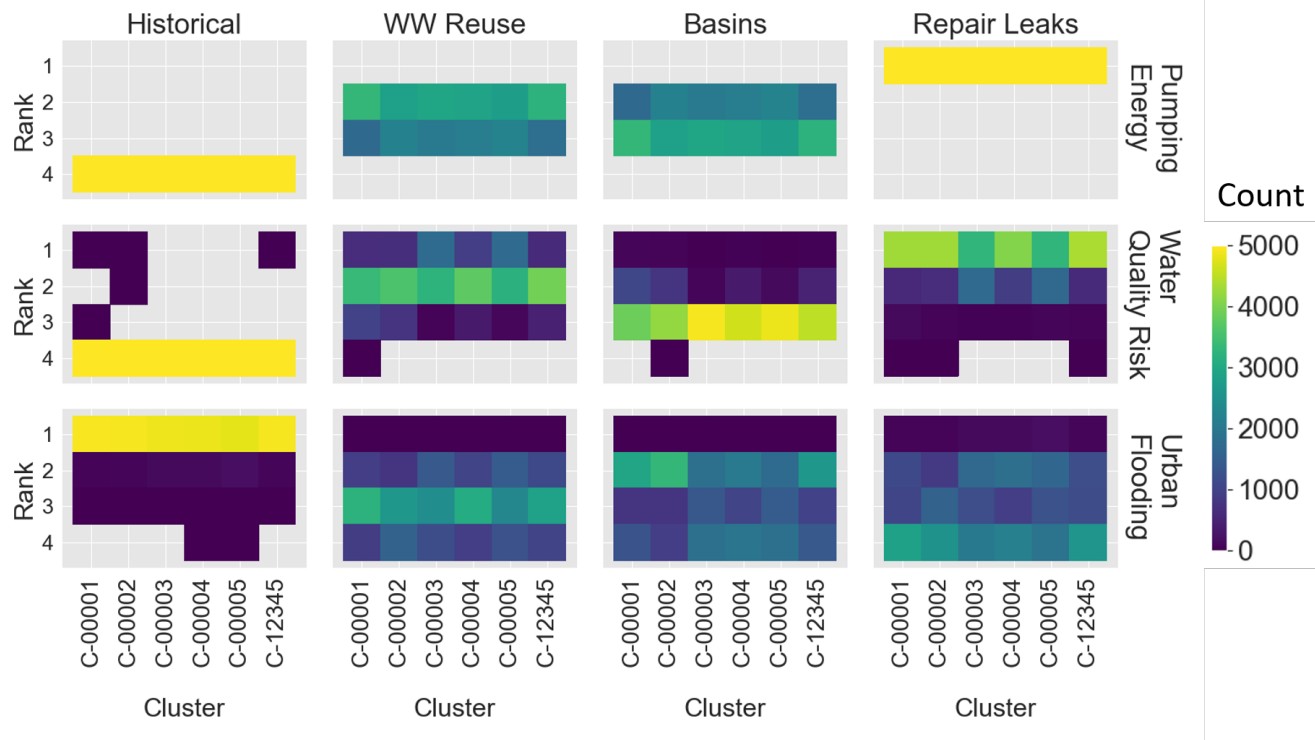

**Figure 7.** Alternative performance across the observation cluster parameter sets shown as heatmaps of the count of sets where the alternative performance was ranked as (1) best to (4) worst. Within each heatmap, the rows are the rank and the columns are the cluster behavioral parameter sets. The subplots are organized by the three management objectives as the rows and the aquifer management alternatives as columns.

have less difference in the number of parameter sets where the repair leaks alternative ranks 1st and the wastewater reuse alternative 2nd when compared to the other cluster behavioral parameter sets (C-00001, C-00002, C-00004, C-12345). Finally, in the urban flooding objective, the best performing alternative is the historical alternative in the vast majority of the parameter sets across all cluster behavioral parameter sets. This is expected given that the urban flooding objective measures groundwater

mounding in the model, and since the remaining three management alternatives all increase recharge in the model, the status quo alternative experiences the least amount of mounding. However, the relative ranking between the other three alternatives is much less clear, particularly in the C-00003 and C-00005 cluster behavioral parameter sets.

Here, it is apparent that the choice of observations by spatial clusters would have a minimal effect on decision-making, making this type of comparison of the alternatives robust across behavioral parameter sets chosen using observations from

many different regions within the model area. This reveals two main points: first, the apparent agreement between sensitivities of performance to parameters across the alternatives may indicate relative stability of the performance of alternatives across the cluster behavioral parameter sets, even though parameter sensitivities are not consistent across those same sets. Second,





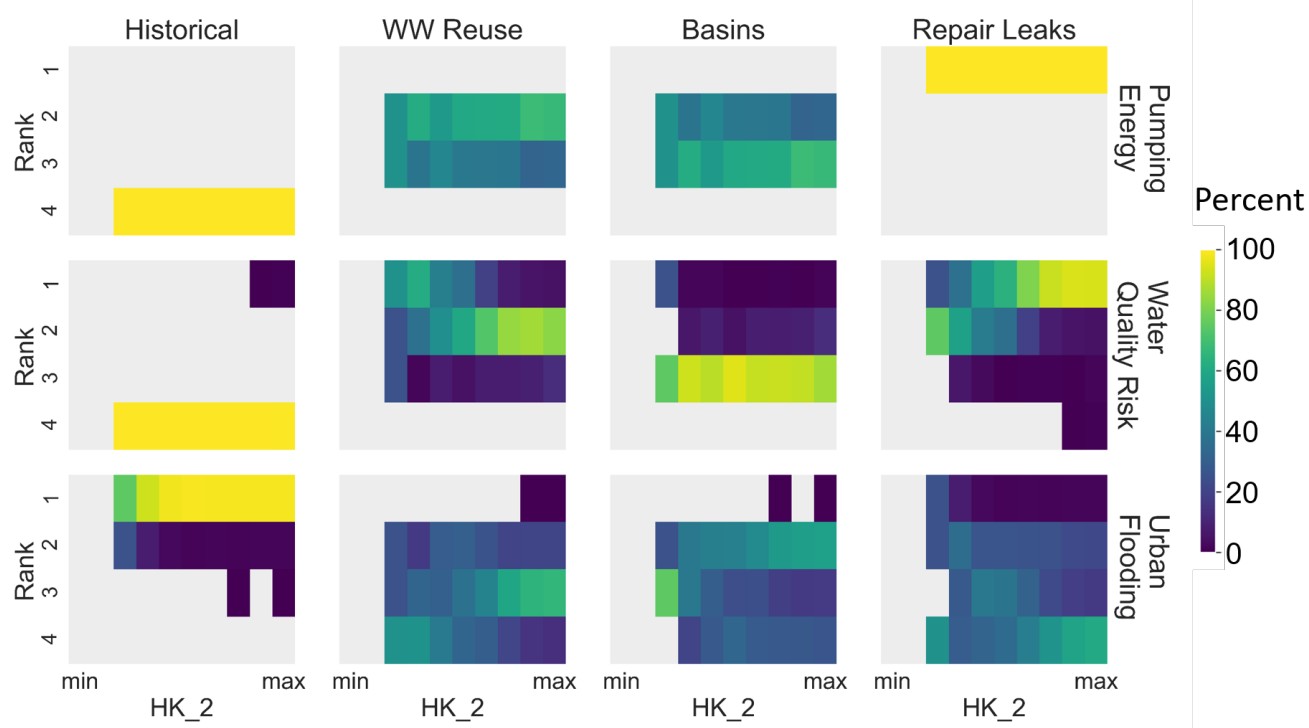

**Figure 8.** Alternative performance across the parameter range of the alluvial hydraulic conductivity (one of the most sensitive parameters) shown as heatmaps of the count of sets where the alternative performance was ranked as (1) best to (4) worst. Within each heatmap, the rows are the rank and the columns are the parameter value from minimum (1.00E-1) to maximum (1.00E+2). The subplots are organized by the three management objectives as the rows and the aquifer management alternatives as columns.

the comparison of rankings across the observational clusters may not capture the full interplay of absolute performance under observational uncertainties.

Next, in Figure 8 we compare the ranking across one of the most sensitive parameters, the hydraulic conductivity of the alluvial formation [HK2], looking only at the behavioral parameter sets chosen using the C-12345 (full) observation cluster. Similar to the comparison across observation clusters, the ranking of the management alternatives across the range of parameter values is stable for the pumping energy objective. The wastewater treatment and infiltration basin alternatives show a roughly even split between the 2nd and 3rd ranking. However, in the other two objectives, the ranking changes depending on the value

of the alluvial hydraulic conductivity. This is particularly apparent in the water quality risk objective, although it also occurs to a lesser degree in the urban flooding objective. Notably, the repair leaks alternative ranks 1st in the water quality risk objective except at lower values of the parameter range, where the wastewater reuse objective is preferred. There are many competing factors that could contribute to this outcome. For example, lower hydraulic conductivity in the alluvial aquifer would indicate higher groundwater retention and could thus favor parameter sets with lower urban leak and total water use values, to reduce



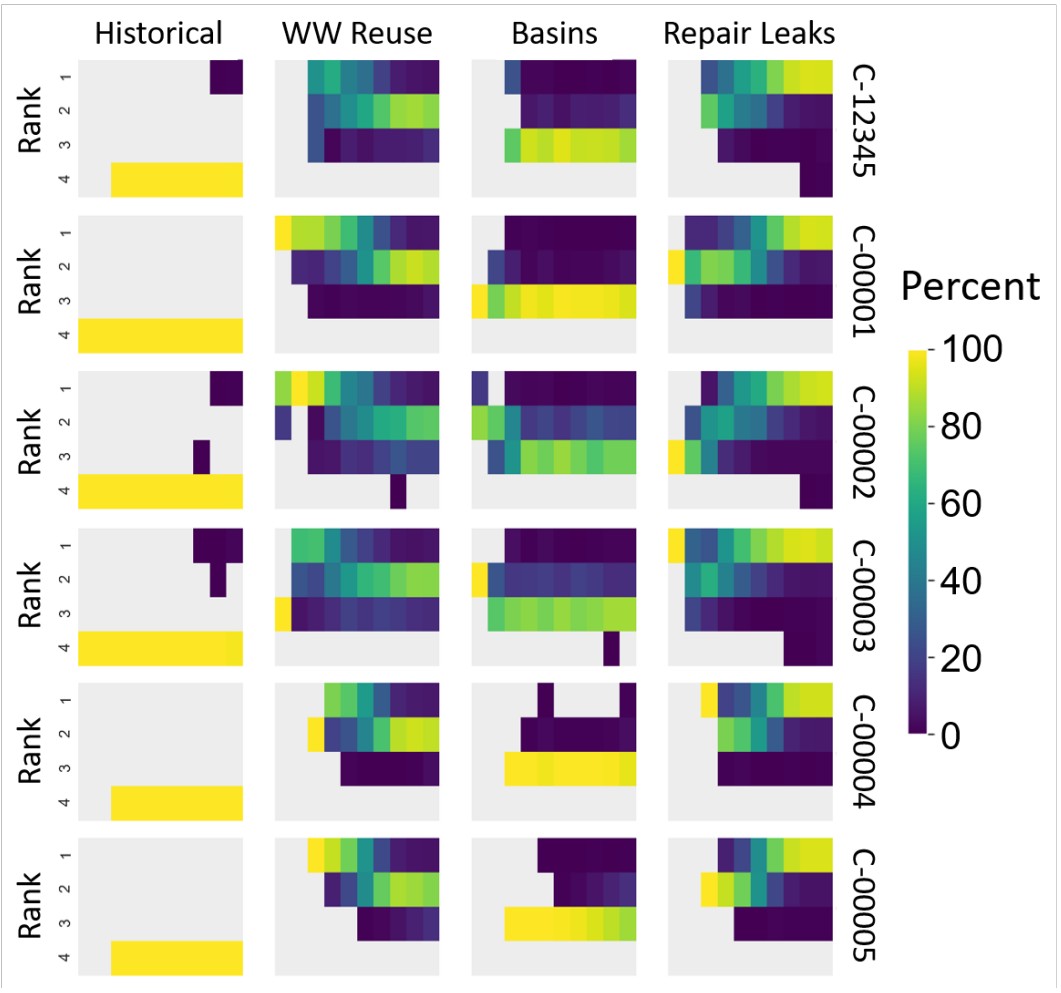

**Figure 9.** Alternative performance in the water quality objective. Shown as heatmaps of the count of parameter sets where the alternative performance was ranked as (1) best to (4) worst. Within each heatmap, the rows are the rank and the columns are the parameter value from minimum (1.00E-1) to maximum (1.00E+2). The subplots are organized by the observation cluster used for behavioral parameter set selection as the rows and the aquifer management alternatives as columns.

model error by avoiding local mounding and cones of depression. In those cases, the wastewater treatment alternative would increase groundwater recharge more than the repair leak alternative, and thus improve groundwater levels within the clay layer that influences water quality risk in the basin. Additionally, some of the fluctuations in the ranking result from the sample counts in each bin after the behavioral parameter set filtering. For example, the lowest bin in the water quality risk objective of the historical and infiltration basin alternatives shows a large difference because the sample count is low. In this case, a change
in the bin size could change the relationship between the parameter values and the alternative ranking.



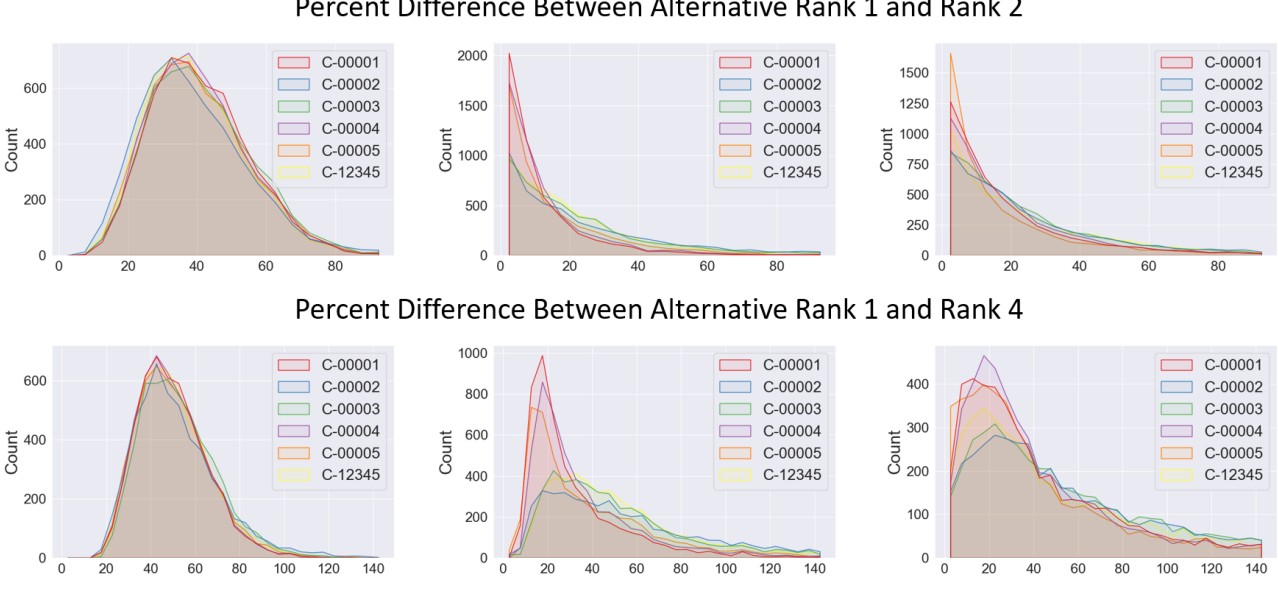

**Figure 10.** The difficulty of the decision represented by the relative performance of the alternatives within the samples evaluated for each objective (columns). The top row shows the distribution of the percent difference in each sample between the 1st and 2nd ranked alternatives within the cluster datasets. The bottom row shows the distribution of the percent difference in each sample between the 1st and 4th ranked alternatives within the cluster datasets.

Finally, Figure 9 shows the combined effects of the observation and parameter uncertainty on alternative performance in the water quality risk objective. Here, it is apparent that the observation cluster choice has an effect on the ranking patterns of the management alternatives across the parameter range. While the pattern of favoring the wastewater reuse alternative at the lower alluvial hydraulic conductivity values and the repair leaks at the higher conductivity values is consistent across all the

observation cluster behavioral parameter sets, the point along the parameter values at which this occurs changes between the clusters used to evaluate model error. There is even a case, at low alluvial hydraulic conductivity in the C-00002 set, where the wastewater reuse, infiltration basins, and repair leaks alternatives are ranked 1st, 2nd, and 3rd, respectively, in contrast with the findings from Figure 7 and, to some extent, Figure 8. This makes clear the importance of evaluating the coupled effects of multiple types of endogenous uncertainties on management outcomes in concert, rather than in isolation.

To visualize the effects of the cluster behavioral parameter set on the difficulty of the decision, Figure 10 shows the distributions of the percent differences between the 1st and 2nd ranked alternatives in each sample (row 1) and between the best (1st) and worst (4th) ranked alternatives in each sample (row 2) for each cluster behavioral parameter set. In this figure, a distribution that is clustered near the origin of the graph indicates a more difficult decision because the percent difference between the objective values of each of the alternatives is smaller.





In the pumping energy objective, the minimal differences in the distributions confirm the conclusions from Figure 7, that the alternative rankings are not affected by which cluster behavioral parameter set was used for calibration. However, in the water quality risk objective, and to a lesser extent in the urban flooding objective, the cluster behavioral parameter set has an effect on the distribution of the percent difference between the 1st and 2nd ranked alternatives as well as the best and worst ranked alternatives. In the water quality risk objective, C-00001, C-00004, and C-00005 show more instances of difficult
decisions. These same cluster behavioral parameter sets also showed more difficult decisions in the urban flooding objective. This indicates that the availability of observational data could contribute to changes in the decision-making process when using the urban flooding and water quality risk objectives in this system.

### 3.4   Limitations and Future Work

Uncertainty analyses face limitations from model complexity and the sample size needed to capture multiple interacting forms
of uncertainty. This study can be extended in several ways to address the challenge of propagating uncertainties throughout the groundwater infrastructure modeling and planning process. For example, this study did not consider multiple model structures and their effects on objective sensitivity and alternative ranking. Such changes could include varying representations of model geology and feedbacks after the implementation of management alternatives. Similarly, the use of observation clusters to reveal spatially dependent sensitivities may obscure the role of outlier observations on parameter sensitivity. Future work could
identify the individual observations that contribute the most to sensitivity in each objective across the various parameters to understand better the limitations of the available observations, as has been achieved in other local sensitivity analysis approaches (Poeter et al., 2014; Matott, 2005; Tonkin et al., 2007).

     As previous studies have applied space-time optimization for groundwater monitoring networks to reduce the variance of water quality estimates, future studies can apply similar techniques combined with the $\delta$ sensitivity measure of groundwater
management objectives to determine optimal sampling locations. Additionally, uncertainty in the proper weighting of observations could be simulated using Monte Carlo selection of weights. Finally, clusters were chosen spatially in this study to simulate over-representation of certain areas in monitoring; however, future research may compare clusters based on physical properties such as land use and geologic formation, or other factors, such as the time period or the agency collecting the data. Similarly, bootstrapping or random selection instead of clustering could reveal the outsized influence of certain individual observations
on parameter calibration and decision-making.

     Additionally, while this study investigates parameter sensitivity and the effects of parameter uncertainty on ranking decisions, it does not explicitly quantify the relationship between the two. The results do not show a clear relationship between the magnitude of the sensitivity of the objectives to changes in the parameters. However, relative differences in the sensitivities of the objectives under different management alternatives may play a role in the alternative ranking. This relationship could
be further investigated by developing a metric to capture the fluctuations in ranking driven by each parameter, to be compared with the differences in sensitivity of a given parameter under each alternative. Similarly, while this study considers the impact of coupled uncertainties on three different management objectives, future work could implement a multiobjective approach evaluating Pareto optimality to consider all three objectives simultaneously.





Finally, the implementation of groundwater recharge alternatives could be modified to improve the accuracy of the simula-
tions. One option would be to include costs of the management alternatives as either an additional objective or as a constraint
to the implementation. Similarly, combinations of the various management alternatives or varying degrees of implementation
may give further multi-objective benefits beyond those of each management alternative implemented individually.

## 4 Conclusions

In this study, we explore how observation and parameter uncertainty propagate through a hydrogeologic model to influence
the ranking of decision alternatives. Using global sensitivity analysis and evaluation of aquifer management objectives across
behavioral parameter sets filtered from a global sample, we evaluate how physical properties of the model and choice of
observations for calibration can lead to variations in decision-relevant model outputs. We find that metrics that are generally
used to determine predictive ability, such as the sum of squared weighted residuals, are not necessarily aligned with the
decision-making applications for which models are applied. The management objective values in the behavioral parameter
samples show a much greater range of sensitivity than those demonstrated by the model error. This underlines the importance
of carrying through sensitivity analyses to the decision-making stage of the modeling process, beyond just the parameter
calibration stage.

Additionally, results show that observational uncertainty plays a much larger role in the sensitivity of the objectives than
the management alternatives themselves. This suggests that the performance of the system with respect to the management
objectives is minimally affected by the choice of alternative when compared to the variability produced by endogenous model
uncertainties. Under certain conditions, the relative performance of the alternatives under some of the objectives is consistent
across many combinations of parameters and observation clusters–particularly for the pumping energy objective. This confirms
that the performance of the demand management represented by the leak repair alternative is robust across many realizations
of uncertainty.

The choice of observations shows a minimal effect on decision-making, with almost no differences in alternative ranking
between the behavioral parameter sets. In contrast, the ranking of the leak repair and wastewater reuse alternatives showed
fluctuations in ranking across the range of one of the most sensitive model parameters, the alluvial hydraulic conductivity.
However, when combined with the parameter uncertainty, the observational uncertainty does contribute to greater fluctuations
in alternative ranking. This makes clear the importance of evaluating the coupled effects of multiple types of endogenous
uncertainties on management outcomes in concert, rather than in isolation.

Finally, the selection of alternatives becomes more or less difficult according to the relative performance of management
objectives. Specifically, the distribution of the difficulty metric in each of the objectives changes based on the observation clus-
ter used to select the behavioral parameter sets. These methods could be leveraged to determine which additional observations
would help to more easily identify the best performing alternative under multiple management objectives. This study highlights
the importance of understanding how the uncertain parameters of a physical model and their interactions with the observations
used to calibrate them can affect water supply planning decisions in densely populated urban areas.





*Code and data availability.* The model with input datasets, observations, results, and postprocessing scripts are available in a GitHub repository at https://github.com/mrlmautner/UrbanGW/tree/sensitivityanalysis (DOI: 10.5281/zenodo.4706103).

*Author contributions.* Marina R.L. Mautner: Conceptualization, Methodology, Software, Data curation, Formal analysis, Writing - original
draft. Laura Foglia: Conceptualization, Writing - review and editing, Supervision. Jonathan D. Herman: Conceptualization, Writing - review and editing, Supervision.

*Competing interests.* The authors declare that they have no known competing financial interests or personal relationships that could have appeared to influence the work reported in this paper.

*Acknowledgements.* This work has been supported in part by the Ford Foundation Predoctoral Fellowship Program of the National Academies
of Science, Engineering, and Medicine. Research trips to Mexico City were funded in part by the University of California Davis Henry A. Jastro Graduate Research Award. We thank the Organismo de Cuencas: Aguas del Valle de México (OCAVM) of the National Water Commission (CONAGUA) of Mexico and the Instituto de Geofísica of the Universidad Nacional Autónoma de México (UNAM) for providing pumping and observation data, the conceptual groundwater model, and input on potential management alternatives.





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
