# Peer review of "Coupled effects of observation and parameter uncertainty on urban groundwater infrastructure decisions"

_Hydrology and Earth System Sciences, 2021_

## Author Response (AR1)

We would like to thank the editor and reviewers for their feedback. Our responses to each point are shown below in regular font, while the reviewer comments are italicized in blue. Excerpts from the manuscript are shown indented in quotations with changes in red.

*Reviewer 1: This study employs global sensitivity analysis to propagate both observation and parametric uncertainty through a model to understand their effects on decision-making for the system. I found the paper interesting and generally well articulated. I had a hard time finding the supplementary figures (maybe they're somewhere in the preprint page I can't see?).*

Thank you very much for your comments, they have helped with clarity and improved the manuscript.

*The challenge the authors attempt to tackle is very complex and the dimensionality of experiment is very large—I don't believe I've seen other papers tackling uncertainty in such a multi-pronged manner (many parameters, many observation locations, many strategies, many objectives). Approaches like the one demostrated in this paper are valuable contributions in paving the way and setting a standard for others.*

We have added the following to the text to the first paragraph to underscore the importance of uncertainty affecting decisions,

> "Further, the effects of endogenous model uncertainties on model error may be different from their effects on the ranking of alternatives, and therefore on decision making. This difference has been largely understudied and is the focus of this paper."

*My comments are all minor suggestions and are detailed below.*

*Introduction is very well written. Goals are articulated clearly and it made me interested to keep reading. One suggestion would be to potentially drive home the difference between uncertainty affecting errors and uncertainty affecting decisions even more. I see the point and I agree with it, but feel like a lot of people don't and it would be nice to have it emphasized a little more.*

*Lines 102-105: Could you include a little more explanation about how these alternatives were selected? I also think stating the general goals/objectives of these management alternatives first might be better.*

The management alternatives were developed and described in greater detail in Mautner et al. (2020). We have changed the text as follows,

> "Four management alternatives designed to increase groundwater recharge within the basin while avoiding flooding are drawn from Mautner et al. (2020). The alternatives were chosen based on conversations with local practitioners and previous modeling efforts. The alternatives are: the implementation of spatially distributed infiltration basins, demand management through repair of leaks in the water supply network, injection of treated wastewater at existing wastewater treatment plants, and the status quo historical alternative."

*Lines 107-108: Can you clarify what this sentence means? How were they adapted?*

The management objectives are drawn from the previous study. However, the calculation of the objectives was modified to avoid outlier values that would occur when parameter combinations led to high quantities of model error. For example, dry cells are represented in the model as large negative values for groundwater head and thus lead to unrealistic values for energy use based on the elevation of groundwater head. These outlier values of the objectives led to difficulties in characterizing the

sensitivity of those objectives to the parameters. Thus, a minimum value for groundwater head was set for each well based on the model cell and layer depth. We have modified the text as follows,

> "The management objectives evaluated are drawn from Mautner et al. (2020), modified to avoid outlier values that would occur when parameter combinations led to high quantities of model error that would affect the sensitivity analysis."

*Line 166: "it is necessary to simplifying" – maybe missing a word there?*

Correct, it should read,

> "While perfect monitoring and representation is the ideal, in reality, simplifying assumptions must be calibrated to create models that can better inform policy and management."

*Lines 168-170: The first part of this sentence needs to be edited I think: "changes in […] can lead to […]"*

We agree, it is more clear as follows,

> "Changes in the observations used to evaluate error can lead to differences in the behavioral parameter sets that are chosen as the best performing simulations."

*Lines 255-265: This is cool analysis and finding. My first reaction was to find out about this parameter's interactions with others, across observation clusters/objectives. I was also curious to look at the supplemental figures referenced, but I couldn't find the supplemental material in the system (not the authors' fault, maybe there's something obvious I am missing?).*

We have attached the supplemental figures to this reply and will include with the manuscript.

*Fig. 5-6 & discussion: I know the interpretation of δ values is context specific and relative, but could you maybe add a sentence or two to give the reader some intuition of what is a 'high'/decision-relevant δ value in such case studies?*

We have added the following text at the beginning of Section 3.2 following the first sentence introducing Figure 5,

> "The value of δ can range from 0, indicating that the output is independent of the parameter in question, to 1. There is not a standard value for δ that is considered to be highly sensitive because parameter sensitivities should be evaluated in relation to each other and in the context of each case study. Based on the sensitivity values for this system, we consider a $\delta$ of roughly 0.2 and above to be highly sensitive."

*Reviewer 2: The study analyses the joint impact of parametric and observational uncertainties on decision objectives of a groundwater resource management problem. The manuscript makes a novel contribution towards understanding the impact of endogenous uncertainties on ranking of alternatives. Overall, the manuscript is well-written with well designed figures. Listed below are some minor corrections and a few recommendations that may help further improve the manuscript.*

Thank you for your detailed comments, we have responded to each point below.

*1. Abstract uses the term 'external' and 'endogenous' uncertainties. This interpretation of this may vary widely across audience so may be better to rephrase by giving examples of what are 'external' uncertainties briefly.*

We agree, the text has been amended as follows,

> "While the impacts of external uncertainties, such as climate and population growth, have been widely studied, there is limited understanding of how decision support is altered by endogenous uncertainties arising from model parameters and observations used for calibration."

*2. Line 11-12 'model errors are not …' this is not surprising as previous studies have shown that across different choice of objective functions, sensitivity of inputs may vary widely. It may be useful to rephrase this or make this specific say 'statistical sqaured error metrics based on piezometric head'.*

We agree, the text has been amended as follows,

> "Error metrics (i.e., squared residuals of piezometric head) are not necessarily controlled by the same parameters as the head based objectives needed for decision-making."

*3. Line 40, possibly missing 'and' after 'system'.*

Yes, this is correct. Updated text:

> "In hydrogeologic models, endogenous uncertainty is contributed by model parameters describing natural and human components of the system, and the set of historical observations used to calibrate or constrain the parameters."

*4. Line 57 on Etter et al. (2018) is not clear, consider rephrasing.*

Upon consideration, the sentence beginning "Etter et al. (2018)…" has been removed in the revision.

*5. Lines 116-118: This part may benefit from first introducing the rationale for a flood risk objective in a groundwater management context before moving on to the definition. Similarly Lines 112-113 state that ensuring hydraulic head 'above' the confining layer reduces the water quality risk, but the immediate next sentence suggests that keeping the levels 'below' is better. Please clarify.*

We agree that this could be more clear, we have added the following text to provide this context,

> "In conflict with the previous two objectives, certain parts of the city lie in areas that are affected by seasonal flooding resulting from medium-term groundwater mounding, which is particularly damaging in urban areas. To take into account these possible negative effects from increasing groundwater head within the valley, the urban flood risk (F) is the sum of the urban

area in cells with groundwater mounding (a) divided by the total urban area in the model (A) during the time periods (t) in the last year of the model period."

*6. Lines 134: useful information on run time, may add the run time for a single model run.*

The model would take between 1 and 30 minutes, with most on the order of 5-8 minutes depending on the processor itself and the parameter combinations. We have added the following line to the text,

> "A single model run is on the order of 5 minutes, depending on the combination of parameters and the processor speed."

*7. Line 146: one point per well implies one observation at an t in the results section to show that the two choices perhaps yield similar outcomes.*

This comment may have been cropped, but we interpret it to refer to lack of clarity on the number of observations used and how that may affect the results. We used all the available observations, which ranged from 1 to 29 annual observations during the 30 year model period, depending on the well. The text has updated as follows,

> "Well observations vary from 1 to 29 data points per well over the 30 year model period, with a maximum of one observation per year."

[Comment numbers skip from 7 to 12]

*12. Line 176-177: using across twice in the sentence makes it harder to understand*

We agree, the text has been amended as follows,

> "The sensitivity of the management objectives across the parameter space with respect to both management alternatives and cluster behavioral parameter sets indicates the variability of uncertainty with respect to individual physical model parameters."

*13. While the overall methodology is clear, more details related to the global sensitivity analysis may help in improving the reproducibility of this analysis. For example, if possible the equation for calculating sensitivity index (delta), may be useful to include in a generalized form. The specific application of this general equation can be discussed for the case study. For example, using a consistent term Y for output on which sensitivity is estimated and then X for the possible input terms and then elaborated with Y and X are for this particular experiment.*

The equation for delta and the accompanying text description have been added as follows,

> "$\delta_i = \frac{1}{2} E_{X_i} \left[ \int \left| d\mu_Y - d\mu_{Y|X_i} \right| \right]$ ,
>
> where the moment independent sensitivity indicator of parameter $X_i$ with respect to the output $Y$ ($\delta_i$) represents the normalized expected shift in the distribution of $Y$, as a function of $\mu_Y$ and $\mu_{Y|X_i}$ , the unconditional and conditional measures of $Y$, respectively. In this study, the parameters ($X_i$) are the 33 parameters shown in Table 1 and the output ($Y$) are the three management objectives described in Equations 1-3."

We have also updated the objective equations (Equations 1-3), so that they read $Y_E$, $Y_W$, and $Y_F$.

*14. Line 201 is quite similar to Line 199 – is it possible to rephrase this or present this as bullet list?*

We have rewritten the paragraph in question to be more clear as follows:

> "We evaluate the model results to visualize changes in ranking according to three types of comparisons using sets of heatmaps that summarize the ranking of the alternatives across all the parameter sets. The comparisons evaluated are: (1) all three objectives across the observation cluster behavioral sets; (2) all three objectives for observation cluster behavioral set C-12345 across the range of the alluvial hydraulic conductivity parameter [HK2]; and (3) the water quality objective ($Y_W$) across the observation cluster behavioral sets and parameter HK2, simultaneously.
>
> In all three comparisons, the first step is to rank the alternatives in relation to each other according to the objective(s) from lowest (1) to highest (4) in each individual parameter set. Then, the ranking data for all the parameter sets in each comparison are summarized as follows:
>
> **Three objectives across observation cluster behavioral sets -** The count of rankings for each alternative. Each column (cluster) in each objective row will sum to 5,000.
>
> **Three objectives for C-12345 across parameter HK2 -** The 5,000 sample set is separated into ten bins along the parameter value range from Table 1. The ranking count in each bin is divided by the total number of parameter samples in each bin to allow for direct comparison across all bins. This is necessary because the distribution of behavioral parameters can be non-uniform. Each column (parameter value bin) in each objective row will sum to 100%, or null if there are no parameter sets in that bin.
>
> $Y_W$ **across the observation cluster behavioral sets and HK2, simultaneously -** Same as in the previous comparison, but for only the water quality objective. This is repeated for the remaining observation cluster behavioral sets (C-00001 to C-00005). Each column (parameter value bin) in each cluster row will sum to 100%, or null if there are no parameter sets in that bin."

*15. Figure 4, units of 'error' may be mentioned. The lower limit is quite high (10^5), not sure whether this range of error is expected.*

This value is the sum of squared weighted residuals, meaning that it is in units of m$^2$ and is expected to be large because there are a total of 8,181 observations. Missing simulated values for an observation are by default represented in the model as -1E6, which can be a result of a dry cell or an incomplete model run based on incompatible parameter values. Therefore, the squared residual would be on the order of 1E12, leading to the highest values seen in Figure 4. We have added the units of m$^2$ to the caption to clarify this point.

*16. Figure 5: using bootstrapping to check the convergence of the sensitivity index or to identify a confidence range on the indices might be helpful in discussing which differences are significant.*

We agree that this will help improve the analysis. The 95% confidence interval for each sensitivity value has been added as a red line over each bar and the caption was updated to indicate this change. The confidence intervals have also been included in the relevant supplemental figures. In general, we find that the differences in parameter sensitivity discussed in the text are significant (i.e., 95% intervals do not overlap), which supports the convergence of the sensitivity analysis as recommended by the reviewer. The following text was added to the caption,

> "The bootstrapped 95% confidence interval for each sensitivity value is shown as a red line."

[Figure]

Parameter Sensitivity for Low Error Sample 0.05: Pumping Energy Objective

17. *Figure 6 caption states '33' parameters but shows 8.*

You are correct, this has been fixed as follows,

> "$\delta$ sensitivity of the error metric and three management objectives (rows) according to the 5,000 filtered samples for the 8 model parameters (columns) with the largest differences in sensitivity between clusters for the historical management alternative."